# Helminth Interactions with Bacteria in the Host Gut Are Essential for Its Immunomodulatory Effect

**DOI:** 10.3390/microorganisms9020226

**Published:** 2021-01-22

**Authors:** Milan Jirků, Zuzana Lhotská, Lucia Frgelecová, Oldřiška Kadlecová, Klára Judita Petrželková, Evan Morien, Kateřina Jirků-Pomajbíková

**Affiliations:** 1Biology Centre, Czech Academy of Sciences, Institute of Parasitology, Branišovská 31, 370 05 České Budějovice, Czech Republic; zuzana.lhotska@paru.cas.cz (Z.L.); hlozkova@paru.cas.cz (O.K.); petrzelkova@ivb.cz (K.J.P.); 2Department of Medical Biology, Faculty of Science, University of South Bohemia, Branišovská 31, 370 05 České Budějovice, Czech Republic; 3Department of Pathology and Parasitology, University of Veterinary and Pharmaceutical Sciences Brno, Palackého tř. 1/3, 612 42 Brno, Czech Republic; frgelecoval@vfu.cz; 4Institute of Vertebrate Biology, Czech Academy of Sciences, Květná, 8603 65 Brno, Czech Republic; 5Department of Botany, University of British Columbia, 3156-6270 University Blvd., Vancouver, BC V6T 1Z4, Canada; evan.morien@gmail.com

**Keywords:** *Hymenolepis diminuta*, helminth, intestinal inflammation, colitis, bacterial microbiota, microbial changes, immune markers

## Abstract

Colonization by the benign tapeworm, *Hymenolepis diminuta*, has been associated with a reduction in intestinal inflammation and changes in bacterial microbiota. However, the role of microbiota in the tapeworm anti-inflammatory effect is not yet clear, and the aim of this study was to determine whether disruption of the microflora during worm colonization can affect the course of intestinal inflammation. We added a phase for disrupting the intestinal microbiota using antibiotics to the experimental design for which we previously demonstrated the protective effect of *H. diminuta*. We monitored the immunological markers, clinical parameters, bacterial microbiota, and histological changes in the colon of rats. After a combination of colonization, antibiotics, and colitis induction, we had four differently affected experimental groups. We observed a different course of the immune response in each group, but no protective effect was found. Rats treated with colonization and antibiotics showed a strong induction of the Th2 response as well as a significant change in microbial diversity. The microbial results also revealed differences in the richness and abundance of some bacterial taxa, influenced by various factors. Our data suggest that interactions between the tapeworm and bacteria may have a major impact on its protective effect.

## 1. Introduction

Due to coevolution, the vertebrate intestine is a complex and dynamic ecosystem underlined by a myriad of interactions among a variety of organisms; for example, between bacteria and helminths [1,2]. A growing body of research suggests that the tripartite partnership (host–helminth–bacteria) has resulted in complex adaptations that have shaped and continue to shape host physiology. As a result, homeostasis in vertebrates, including humans, is likely to require the presence of both commensal microbiota and macrobiota, i.e., helminths [3]. Helminths are potent manipulators of the host immune system [4], and a number of studies have uncovered their interactions with the intestinal bacterial community (e.g., [5,6]). Accumulating evidence supports the theory that an absence of worms in the intestinal ecosystem can lead to a dysregulated immune system and subsequent dysbiosis. This can escalate harmful inflammatory reactions, contributing to a number of disease states [2,7,8].

The ability of helminths to redirect the host immune response to suppress the inflammatory mechanisms that are responsible for their direct elimination has led to the development of a novel biological therapeutic approach called helminth therapy [9]. Despite the large body of literature describing the mechanisms by which helminths modulate the immune system [10,11,12], the interactions among host, worm, and bacteria have rarely been investigated. It has been shown that the bacterial component itself is able to modulate the host immune system [13,14]. Furthermore, while bacterial dysbiosis can lead to various chronic inflammatory diseases [15], enriching it, on the other hand, can reduce inflammation in the host; for example, by adding beneficial bacteria, such as probiotics [16,17], or using fecal microbial transplantation [18]. Helminths and bacteria that inhabit the gastrointestinal tract have lived in close proximity throughout evolution, so their symbiotic action on the host organism can be expected [6,19,20]. One piece of evidence supporting the importance of this interaction may be the beneficial effect of helminth-modified bacteria on airway inflammation in animals (the bacteria were transferred from colonized to non-colonized animals) [21]. However, it remains to be determined to what extent the bacterial microflora is necessary for the anti-inflammatory action of intestinal therapeutic worms in the human body, and whether changes in bacterial diversity or composition under helminth colonization pressure may alter the effectiveness of helminth therapy.

Some animal model studies that have focused on the relationships between intestinal microbiota and macrobiota have revealed that helminths can alter their surroundings in the host digestive tract, often resulting in a restructuring of bacterial communities [22]. For example, colonizations with the mice nematodes *Heligmosomoides polygyrus* and *Trichuris muris* were associated with changes in the microbial diversity and in the abundance of some bacterial groups [21,23,24,25,26,27]. However, these mice studies report inconsistent results in microbial diversity, which is sometimes increased in the presence of a worm but can also be decreased. In contrast, a study on macaques with inflammatory bowel disease showed more straightforward results, with a shift in bacterial composition during *Trichuris trichiura* colonization toward the conditions observed in healthy animals [28]. Other studies have shown differences in the abundance of some bacterial taxa in pigs colonized with *Trichuris suis* [29,30]. Similarly, studies on human populations have revealed variable results for the changes to intestinal microbiota associated with helminth colonization. While some found differences only in microbiota diversity [27,31,32], others found significant shifts towards increased diversity as well as compositional changes [33,34]. Understanding the interactions between helminth colonization and intestinal bacterial microbiota appears to be critical for the successful implementation of helminth therapy.

The tapeworm *Hymenolepis diminuta* appears to be an optimal candidate for the purposes of helminth therapy. Compared to other helminths that have been used, it fully meets all eligibility criteria, including easy and inexpensive cultivation under laboratory conditions [9,35]. So far, *H. diminuta* has mainly been tested in animals, and the beneficial effect of suppressing inflammation has been shown for several disease models, such as Crohn’s disease, rheumatoid arthritis, and an autism-type neurocognitive disorder [36,37,38,39]; however, this effect was not detected in all studies [40,41]. Additionally, reports from the community of so-called self-treaters suggest a high efficacy for *H. diminuta* (more than 60%) in more than 50 different chronic diseases associated with inflammation and also describe minimal side effects [42,43]. Despite differences in the nature of the immune response and the length of colonization among rodent models (rats versus mice), the beneficial effect of *H. diminuta* on intestinal inflammation has been observed in both cases [39,44].

At present, the relationship between long-term colonization with *H. diminuta* and the bacterial component of the gut microbiome has been investigated in three studies of the rat model [39,45,46]. Two of these studies also involved a moderate or severe model of intestinal inflammation. All of these studies showed that *H. diminuta* does not significantly affect alpha diversity of the intestinal microbiota, rather its presence correlates mainly with changes in bacterial composition (beta diversity). At the taxonomic level, they confirmed changes in the Firmicutes group with a putative bloom of Clostridiales [45] or a drop in *Lactobacillus* [39]. A recent study by Shute et al. [47] investigated the effect of intestinal bacterial microbiota on *H. diminuta* expulsion in a mouse model, but they found no effect on the kinetics of expulsion.

Although the beneficial effect of *H. diminuta* on intestinal inflammation has been confirmed, data on the role of intestinal bacteria in this effect are lacking. Addressing this knowledge gap will advance our understanding of the helminth–bacteria–host interaction and the putative mechanisms by which the helminth may alleviate intestinal inflammation. Therefore, the current study addresses two questions. Firstly, whether disruption of the intestinal microbiota during *H. diminuta* colonization affects the subsequent course of intestinal inflammation, since the bacterial microbiota is known to play a role in intestinal diseases e.g., [48]. Secondly, what changes to the intestinal bacterial community are caused by a combination of various factors (colonization, antibiotics, and colitis) and whether they are reversible. During the study, we also monitored the pro-inflammatory and anti-inflammatory immune markers and the clinical status of experimental rats. This study follows up on our previous study by Jirků-Pomajbíková et al. [39], in which we showed a reduction in intestinal inflammation during the patent period of *H. diminuta* colonization.

## 2. Material and Methods

### 2.1. Ethics, Experimental Design, and Sampling

Outbred female Wistar rats, 13 weeks of age, were obtained for the experiment from Envigo RMS B.V. (Horst, The Netherlands; the supplier Anlab s.r.o., Prague, Czech Republic). All rats were acclimated to the animal facility before the start of the experiment, housed under controlled conditions, had their health status visually inspected at 24 h intervals during daily routines, and were euthanized by cervical dislocation at the end of the experiment. This study was approved by the Committee on the Ethics of Animal Experiments of the Biology Centre of the Czech Academy of Sciences (České Budějovice, Czech Republic, permit no. 33/2018) and by the Resort Committee of the Czech Academy of Sciences (Prague, Czech Republic) according to strict accordance with Czech legislation (Act No. 166/1999 Coll. on veterinary care and on changes of some related laws, and Act No. 246/1992 Coll. on the protection of animals against cruelty), as well as the legislation of the European Union.

The aim of the experiment was to monitor the effect of several factors, specifically worm colonization, intestinal microbiota disturbance, and intestinal inflammation on the health of rats. The experimental design follows up on our previous results from the study by Jirků-Pomajbíková et al. [39]. As a result of all the procedures used (i.e., colonization, antibiotics, and colitis), there were four differently affected experimental groups at the end of the experiment (Day 31; Figure 1): (i) rats treated with *H. diminuta* and antibiotics (HA); (ii) rats treated with *H. diminuta*, antibiotics, and colitis (HAC); (iii) rats treated with *H. diminuta* and colitis (HC); and (iv) rats treated with antibiotics and colitis (AC). Day 31 in the present experiment corresponds to the seventh day after induction of colitis, when we recorded in our previous study by Jirků-Pomajbíková et al. [39] a reduction in intestinal inflammation during the patent period of *H. diminuta* colonization (i.e., adult worms present). Here, we collected various samples, such as the spleen for evaluating cytokine gene expression and the hindgut for microbial and histological analyses, as well as clinical data (Figure 1). Fecal samples for evaluating microbiome changes during the experiment (i.e., after colonization, after antibiotic administration, and after colitis induction) and clinical data were collected throughout the experiment—on Days 0, 18, 24, and 31 (Figure 1).

Maintenance of the *H. diminuta* culture under laboratory conditions and the establishment of rat colonization is described in Jirků-Pomajbíková et al. [39].

Sampling, antibiotics administration, and colitis induction were performed strictly under inhalation anesthesia using isoflurane (Forane 100 mL, AbbVie s.r.o., Prague, Czech Republic) and anesthesia equipment (Oxygen Concentrator JJAY-10-1.4, Longfian Scitech Co. Ltd., Baoding China; Calibrated Vaporized Marts VIP 3000, Midmark, Dayton, OH, USA). Clinical parameters were collected in a blinded fashion and according to protocols described in Jirků-Pomajbíková et al. [39].

### 2.2. Antibiotics Administration and Colitis Induction

Three types of broad-spectrum antibiotics (neomycin, vancomycin, and ampicillin) and an antifungal agent (amphotericin B) were selected to disrupt the intestinal microbiota according to the literature [49,50,51]. Ampicillin (1 g/L) was administered in drinking water, the rest of the antibiotics (neomycin—100 mg/kg/day; vancomycin—50 mg/kg/day; and amphotericin B—1 mg/kg/day) were administered per rectum using a catheter (10 cm long, 3.3 mm diameter; type Nelaton, Dahlhausen s.r.o., Kuřim, Czech Republic). The antibiotics were administered daily for three consecutive days (Days 22–24) in three groups (Figure 1). Colonized rats received a placebo only (PBS). Subsequently, the severe colitis model was induced using dinitrobenzene sulfonic acid (DNBS) (detailed procedure described in Jirků-Pomajbíková et al. [39]). Rats in group HA (colonized with antibiotic treatment) received the placebo.

### 2.3. Analyses of Immune Markers

To evaluate the relative gene expression of immune markers, we collected a portion of the spleen (approx. 150–250 g) of individual rats, from which total RNA was isolated using a kit, Hybrid-R^TM^ (GeneAll Biotechnology, Seoul, Korea), and then reverse transcribed to cDNA using the High-Capacity RNA-to-cDNA kit (Thermo Fisher Scientific, Waltham, MA, USA). cDNA was subjected to subsequent qPCR analyses using commercially available Taqman gene expression assays (Thermo Fisher Scientific) with rat specific probes for TNFα (amplicon length—92 bp), IL-1β (amplicon length—74 bp), IFNγ (amplicon length—91 bp), IL-6r (amplicon length—95 bp), IL-4 (amplicon length—85 bp), IL-13 (amplicon length—95 bp), Stat6 (amplicon length—100 bp), IL-10 (amplicon length—70 bp), and UBC as housekeeping (amplicon length—88 bp) (Figure 2).The entire procedure and software analyses of gene expression are described in detail in the study by Jirků-Pomajbíková et al. [39]. Differences in the cytokine gene expression among individual experimental groups on Day 31 were assessed by non-parametric Kruskal–Wallis ANOVA with multiple comparisons in Statistica (data analysis software system) version 13 (TIBCO Software Inc., 2018, http://tibco.com); (Figure 2).

### 2.4. Histology

Tissue samples of the hindgut were collected for histopathological examination. The samples were fixed in buffered 10% neutral formalin, dehydrated, embedded in paraffin wax, sectioned on a microtome (HM 430 Sliding Microtome, Thermo Shandon Limited, Waltham, MA, USA) at a thickness of 4 μm, and stained with hematoxylin and eosin (H&E). Histological changes were evaluated using light microscopy (OLYMPUS BX51, Olympus Corporation, Tokyo, Japan).

### 2.5. Microbial DNA Extraction, Amplification, and Sequence Processing

Total DNA was purified using the PSP SPIN stool DNA Plus Kit (Stratec Biomedical, Birkenfield, Germany). 16S ribosomal DNA was amplified using primers targeting the V4 region. The entire procedure for amplification and its conditions as well as primers are described in detail in Jirků-Pomajbíková et al. [39].

Amplicon sequences from 170 samples were demultiplexed in QIIME v1.9 [52] and then trimmed, clipped, and quality-filtered using the Fastx Toolkit [53] to 250 bp with a minimum quality threshold of Q19. We processed the filtered R1 reads into operational taxonomic units (OTUs) using minimum entropy decomposition (MED) [54] with the minimum substantive abundance (-m) parameter set to 250 bp. We assigned taxonomies to the representative sequence for each MED node by matching it to the SILVA 128 [55] database using QIIME [52]. We removed OTUs with mitochondrial or chloroplast assignments and those unassigned. As a noise reducing measure, all read counts of 2 or less per sample were removed. The final filtered OTU table consisted of 2390 unique sequences and 10,027,683 reads. The average read depth per sample was 58,986.

We conducted analyses of the OTU table using R version 3.6.1 [56]. Microbial community composition and differential abundance were analyzed using the Phyloseq [57] and DeSeq2 [58] R packages. Data and subsets were rarefied prior to α-diversity and β-diversity analyses. We quantified α-diversity using the Chao1 index [59], while β-diversity was quantified using the Bray–Curtis dissimilarity index [60]. We tested differences in α-diversity between samples using nonparametric multivariate analysis of variance (PERMANOVA) with the function “adonis” in the R package Vegan [61]. All *p*-values were Benjamini–Hochberg corrected [62]. Microbial visualizations were produced using ggplot2 [63].

## 3. Results

### 3.1. Cytokine Results

Significant differences in the relative gene expression between the experimental groups were observed in all monitored immune markers in the spleen samples collected on the last day of the experiment, i.e., Day 31 (TNFα: H = 17.75, *p* < 0.01; IL1β: H = 17.8, *p* < 0.01; IL6r: H = 22.89, *p* < 0.01; IFNγ: H = 16.88, *p* < 0.01; IL4: H = 18.63, *p* < 0.01; IL13: H = 16.86, *p* < 0.01; IL13: H = 16.86, *p* < 0.01; Stat6: H = 16.95, *p* < 0.01; IL10: H = 22.83, *p* < 0.01; Figure 2).

All colitis-treated groups (HAC, HC, and AC) showed significantly higher gene expression of TNFα in comparison to the HA group, the colonized rats treated with antibiotics (*p* < 0.01 for all comparisons with HA; Figure 2). Surprisingly, we did not observe any difference in the reduction of intestinal inflammation based on TNFα relative gene expression in colonized animals with colitis (HAC and HC) in comparison to the other groups (Figure 2). This differs from the results in our previous study by Jirků-Pomajbíková et al. [39], as there we found an amelioration of intestinal inflammation in the group of rats colonized with *H. diminuta* on the seventh day after colitis induction in the patent period (that day corresponds to Day 31 in this study). Interestingly, in this experiment, no trend of recovery was observed in the HC group (i.e., colonized animals with colitis). The anti-inflammatory effect could have been delayed by stress during antibiotic administration (Days 22–24), when rats in the HC group received a placebo treatment.

The relative gene expression of pro-inflammatory marker IL1β was higher in all groups of rats with colitis in comparison to the HA group, but this difference was only significant for the AC group (*p* < 0.01; Figure 2). IL6r was significantly lower in the HA group as compared to the AC (*p* < 0.05; Figure 2) and HAC (*p* < 0.01; Figure 2) groups. The relative gene expressions of IFNγ, IL4, and STAT6 were significantly higher in the HA group in comparison to all groups with colitis (*p* < 0.01 for all comparisons with HA, with the exception of IL4: AC vs. HA, *p* < 0.05; Figure 2). The relative gene expression of the regulatory cytokine IL10 was significantly lower in the HA group as compared to the AC and HC groups (*p* < 0.01 for both; Figure 2). For IL13, the relative gene expression in the HA group was higher than in the other groups, but the difference was only significant in comparison with the AC (*p* < 0.05; Figure 2) and HC (*p* < 0.01; Figure 2) groups.

### 3.2. Histology Results

Histological analysis of the colon samples from all experimental groups on Day 31 revealed that the most severe intestinal inflammation was in the AC group of rats (antibiotic and colitis-treated; Figure 3A), while the HA group (colonized and antibiotic-treated) experienced the mildest amount of inflammation (Figure 3B). Rats in the AC group had severe active ulcerative colitis (purulent to necrotic colitis) with an inflammatory infiltrate (consisting of lymphocytes, plasma cells, and eosinophils) in the mucosa and submucosal layer. In deep ulceration/necrosis, there was a mixed inflammatory infiltrate with a predominance of neutrophils and fibroplasia at the bottom and numerous bacterial colonies on the surface. The inflammation in some necrotic areas diffused to the muscle layer (Figure 3A). A similar degree of inflammation was found in the HC and HAC groups (Figure 3C,D). Whereas rats in the HC group (colonized with colitis) had moderate inflammation, those in the HAC group (colonized, antibiotic-treated, and colitis) experienced mild inflammation with the addition of goblet cell hyperplasia. The colon of rats in the HA group (colonized and antibiotic-treated) showed mild/focal lymphoplasmacytic inflammatory infiltrate with a predominance of eosinophils in the mucosa and mild reactive hyperplasia of the submucosal lymphoid tissue (Figure 3B).

### 3.3. Microbial Results

Changes in the intestinal microbial diversity were observed throughout the experiment—before colonization (Day 0), after colonization (Day 18), after antibiotic treatment (Day 24), and after colitis induction (Day 31) (Figure 1). As the experiment progressed (Days 0, 18, and 24), we collected fecal samples only, while on Day 31, we also collected samples from the colon and caecum, and then compared the microbial diversity in all three types of samples (Figure 4A,B).

The alpha diversity of the rat intestinal microbiome appeared to be stable in colonized and non-colonized rats on Day 18, as it was comparable with Day 0 (Figure 4A; Appendix A). Significant changes were found on Day 24, i.e., after three consecutive days of antibiotic administration (Figure 4A). Surprisingly, the most significant drop in alpha diversity was observed in the group of rats treated with *H. diminuta* and antibiotics, while in the group of rats treated with antibiotics alone, the change was less profound (Figure 4A; Appendix A). In both these groups, significant compositional changes were recorded on Day 24 (Figure 4B).

On Day 31, after colitis treatment, alpha diversity in all the antibiotic-treated groups (AC, HAC, and HA) rebounded (Figure 4A), but there was a lasting change in microbiota composition (Figure 4B). According to our results, colitis did not have a major effect on the restoration of diversity in the antibiotic-treated groups (AC and HAC vs. HA), and the diversity did not return to its original state in any of these groups (Figure 4B). Worm colonization did not promote a return of the rat gut microbiome to its original diversity and composition in the groups treated with antibiotics (HA), colitis (HC), or both (HAC) (Figure 4B). There was a significant cage effect in the HA group (Figure 4B; more details below).

The proportions of various bacterial taxa present in the rat intestine differed significantly between all groups on Day 31 (Figure 5): (i) group AC—Akkermansiaceae and Bacteroidaceae taxa predominated; (ii) group HAC—a similar trend as the previous group, with the additional predominant taxon Muribaculaceae; (iii) group HC—significant proportions of Lachnospiraceae and Muribaculaceae taxa; and (iv) group HA—the application of antibiotics evidently enhanced the cage effect on the taxonomic composition of intestinal microbiota as the cage mates were always more similar to each other than to the other individuals in their group (for HA1 and HA2 Rikenellaceae predominated; for HA3 and HA4 Lachnospiracea predominated; in the other two cage mate pairs there was a large proportion of Bacteroidaceae, with HA7 and HA8 additionally having a significant proportion of Muribaculacea). We confirmed the trends in taxonomic representation in each group using three types of samples—fecal, colon, and cecum. All sequence data and MiMARKs compliant metadata for this study are available at the European Bioinformatics Institute (accessions PRJEB35048/ERP118040).

### 3.4. Clinical Data

During the experiment, there was no death of an animal due to the application of antibiotics or induction of colitis. Clinically, the groups differed on Day 31 in terms of the ratio of individuals with hematochezia, as well as fecal consistency and weight (Appendix A). We found three or four *H. diminuta* adults in each experimental rat.

## 4. Discussion

The tapeworm *Hymenolepis diminuta* is a benign helminth that has been shown to have a beneficial effect on inflammatory diseases [9]. Recent studies also suggest it interacts with the intestinal bacterial microflora [39,45,47]. However, the relationship between *H. diminuta* and bacteria and its impact on the host organism has not yet been sufficiently elucidated, even in the case of the anti-inflammatory action of *H. diminuta* in the host intestine. Therefore, we focused here on determining the potential role of bacterial microflora in the protective effect of *H. diminuta* on intestinal inflammation, which was found in our previous study [39]. These results also revealed changes in the microflora of the rat intestine during colonization by mature *H. diminuta*, during induction of colitis, and during recovery from intestinal inflammation. Here, we used the same experimental design as our previous study with the additional inclusion of disrupting the bacterial microbiota using broad-spectrum antibiotics before inducing colitis. Sampling and observations were performed on the same experimental day (see Material and Methods for details) when we previously recorded the most significant reduction in intestinal inflammation due to *H. diminuta* colonization [39]. To better understand our results, it is important to note that four different experimental groups were formed in the end of experiment (i.e., AC, HC, HA, and HAC; Figure 1).

Surprisingly, we did not observe any protective effect of *H. diminuta* on intestinal inflammation in either of the groups affected by colonization and colitis (HC and HAC) on Day 31, which contradicts our previous results [39]. The gene expression level of the pro-inflammatory marker TNFα was similar in all the colitis-treated groups (HC, HAC, and AC); thus, it is likely that the prolongation of inflammation is a consequence of stress invoked by additional handling of the rats during antibiotics (HAC and AC groups) or placebo (HC group) administration (Days 21–23). It has been shown that acute and also chronic stress correlates with elevated TNFα levels in rats (e.g., [64]). Our current data suggest that the inclusion of another factor in the experimental design can cause significant changes in the model of intestinal inflammation and preclude comparisons with the results of previous study [39]. In addition, an increased production of TNFα has been linked to gut microbial dysbiosis [13] and to an alteration of the protective effect of worms that is dependent on intestinal bacteria [65]. The association between TNFα and the intestinal microbiota could also explain our previous observation of differences in colitis severity across three independent experiments in non-colonized rats despite consistent experimental conditions [39]. These findings may indicate the importance of bacterial microflora and stress in testing the effect of helminths on intestinal inflammation in animal experimental systems.

In addition to TNFα, an increase of another pro-inflammatory marker was observed in the experimental groups treated with colitis. We found an elevation in the gene expression of IL-1β and IL-6r in the AC group and the latter one also in the HAC group. In the HC rats, there was no activation of any other observed marker. Interestingly, the HA rats showed a strong induction of the anti-inflammatory cytokines IL-4 and IL-13, along with the transcription factor Stat6, all specific for the Th2 immune response. It is worth mentioning that in none of our previous studies did we detect the activity of Th2 markers in the presence of mature *H. diminuta* (i.e., patent period of colonization), but only at the beginning of the prepatent period [39,46]. Hayes et al. [66] demonstrated a similar trend with the induction of a Th2 immune response in mice colonized with *Trichuris muris* and subsequently treated with antibiotics. In a rat model, differences in the activation and course of the Th2 response during *H. diminuta* colonization were reported only with increasing worm number and biomass [67]; however, we used the same number of larvocysts for inoculation of rats across all our studies. In a mouse model (non-permissive hosts), the Th2 immune response is associated with expulsion of the helminth at preadult stages and with an anticolitic effect [44,68,69,70,71].

Differences in inflammatory processes between the colitis-treated groups of rats were also found based on histological examination. Much more progressive inflammation, often towards necrosis, was observed in the groups without the combination of helminth colonization and antibiotics (i.e., AC and HC groups). In contrast, the HAC group showed mild to moderate inflammation, and no inflammation or only rarely focal inflammation was detected in the HA group. A similar trend was found in the clinical data (Appendix A). Unexpectedly, the rats in group AC (treated with antibiotics and colitis) had no alleviation of gut inflammation. However, this may be consistent with the study by Ozkul et al. [72], which found that mice intestinal cells had a high susceptibility to apoptosis after antibiotic and subsequent DSS-colitis treatment. Furthermore, the use of antibiotics alone may have a non-antibiotic effect due to neutrophil apoptosis [73].

From our results, it is difficult to determine exactly what caused the different immune responses in the experimental groups affected by various factors. Some reports have revealed a direct relationship between the activity of different cytokines and changes in the bacterial microbiota [13,17,74,75] or changes in the transcription of immune system genes during helminth colonization [76]. Interestingly, several studies observed a decrease in the activity of pro-inflammatory cytokines in susceptible strains of mice or in mice with chemically induced colitis after oral administration of beneficial bacteria, such as *Lactobacillus fermentum*, *Lactobacillus casei*, or *Akkermansia municiphila* [16,17,65]. Another study by Fricke et al. [77] demonstrated a reduction in the bacterial population in the small intestine of mice due to the induction of a Th2 response after helminth colonization. In addition, interactions between helminths and bacteria in the host gut, occurring at several levels, cause changes in the metabolic pathways, in intestinal wall function, and in the dynamics of mucus production, which may be reflected in the immune response [22]. This suggests that the host immune response (including transcription of immune genes) might be altered not only due to interactions between the helminth and bacteria in the gut, but also due to other factors influencing these coordinated actions. These aspects should be taken into account in the research of helminth therapy.

The microbial results on Day 31 revealed that the differences in the richness and abundance of some bacterial taxa between the rat groups were influenced by various inflammatory factors. Bacteroidales were found in all groups and were often the most abundant taxon, especially in the group of rats treated by antibiotics and colitis (AC). In each group of rats with colitis (AC, HAC, and HC), the abundance of another bacterial group was markedly increased, but the identity of this group was distinct for each treatment (Figure 5). In the groups of rats colonized with *H. diminuta* (HA, HAC, and HC), no common trend in community composition or abundance was found as a consequence of colonization. Significant differences in diversity at the group level were detected in the HA treatment group, which had a strongly induced Th2 response (Figure 4 and Figure 5). This treatment group also displayed differences in the abundances of various taxa at the level of cage mates, especially for representatives of the Bacteroidales (i.e., Bacteroidetes, Rikenellaceae) and Clostridiales (i.e., Lachnospiraceae) groups. This finding is consistent with Ramanan et al. [27], who showed that helminth colonization with induction of the Th2 response alters the ratios of Bacteroidales and Clostridiales groups, which can impact intestinal inflammation. These diverse changes to the microbiota in the HA group correlate with a pronounced elevation of the cytokine IFNγ. Similarly, Holm et al. [25] found an increase in IFNγ secretion after stimulation by helminth derived (*Trichuris muris*) compounds in association with altered diversity and composition of the microbiota. In the case of antibiotics, their administration has been associated with a decrease in IFNγ gene expression [78]. Moreover, the association between microbiota changes and IFNγ secretion has been demonstrated in a large human dataset in the Human Functional Genomics Project [13]. These findings lead us to the conclusion that the changes to microbiota composition in our experiment under the influence of external factors alone—helminth colonization, antibiotic administration, and colitis induction—as well as in combination, had a significant effect on the activity of various pro- and anti-inflammatory markers.

We observed a significant change in the diversity and composition of the bacterial microbiota after three days of antibiotics administration (Day 25), which is similar to previous studies [49,78,79]. Surprisingly, the most significant change in microbial composition was in rats treated by antibiotics and *H. diminuta* colonization, even more than in individuals affected by antibiotics alone (Day 25; Appendix A). In contrast, in colonized rats only, the microbial diversity was relatively stable. Consistent with our data, Shute et al. [47] observed a decrease in diversity after antibiotic administration in individuals colonized with *H. diminuta*, but it is difficult to compare this with our data due to differences in hosts (mice versus rats), experimental design, and sampling days. It has been shown that the effect of antibiotics on the composition of the bacterial community in the gut is long-lasting, impacting metabolic homeostasis [80], and some studies report that this altered gut microbiota returns to near baseline after a month or more [49]. Therefore, longer monitoring in this context is needed.

Previous studies in the rat model were focused on the impact of *H. diminuta* alone on the bacterial microbiota and always demonstrated mainly compositional changes [39,45,46]. In this study, diversity rebounded despite the induction of colitis in some groups (Day 31), yet community composition did not return to the state it was before colitis induction (Day 25) or before antibiotics administration (Day 21)—the microflora was pushed to an alternative state. In the case of colitis, we observed a similar trend as in our previous study by Jirků-Pomajbíková et al. [39]. *Hymenolepis diminuta* in combination with colitis (HC) influenced the bacterial community much less aggressively than in combination with antibiotics (HAC, AC, and HA). The most significant change was detected in the HA group, both during and at the end of the experiment, and was associated with the induction of the Th2 response. Our results suggest the possibility of inducing the protective effect of the adult tapeworm by drastically transforming the gut microbiota, which we performed using broad-spectrum antibiotics, during *H. diminuta* colonization. However, a more beneficial intervention to the microbiota might be enrichment of the microbiome either with probiotic bacteria or fecal microbial transplantation [16,17,18,65], which could also have a beneficial impact on the metabolic homeostasis of the host. Further testing is necessary to understand or resolve this discrepancy. Some recent studies emphasize the importance of the reliance of helminths on bacterial symbionts, both during their development and during host colonization itself [1,19,20]. Thus, it is possible that the protective effect of *H. diminuta*, as well as its reliability, is dependent on the modulation of the host microflora and their symbionts.

## 5. Conclusions

Our findings uncover the intricate interactions of the tripartite partnership within the helminth–microbiome–host complex. It is evident that the ability of the helminth to modulate the immune system depends, at least in part, on the intestinal microflora, and this interaction in turn may impact its protective effect in inflammatory disease. Our results also point to the importance of various factors (such as helminth colonization, antibiotic administration, and colitis induction) might have a significant effect on the changes in gut microbiota alone and subsequently on the activity of various pro- and anti-inflammatory markers. It is very likely that changes in the intestinal microbiome as a result of colonization with the benign tapeworm *H. diminuta* alone are not sufficient to activate a strong and long-lasting anti-inflammatory mechanism, even during intestinal inflammation.

In the future, comprehensive studies focusing on the interactions among helminths, bacteria, and the host organism should consider the aspect of the long-term monitoring of the immune response and intestinal microbiome. As our study showed, a single time point is not sufficient for the interpretation of changes in the immune response and the full pattern of changes in the intestinal community in the host organism under pressure of different factors, including stress. During follow-up investigation, it would be interesting to determine whether the bacterial community in one of the experimental groups returns to its original state faster than in the others and, in the case of the HA group, how long the Th2 immune response persists. It would also be useful to identify the bacterial metabolites that are produced during *H. diminuta* colonization, which may provide better insight into the mechanisms within the helminth–microbiome–host relationship.

## Figures and Tables

**Figure 1 microorganisms-09-00226-f001:**
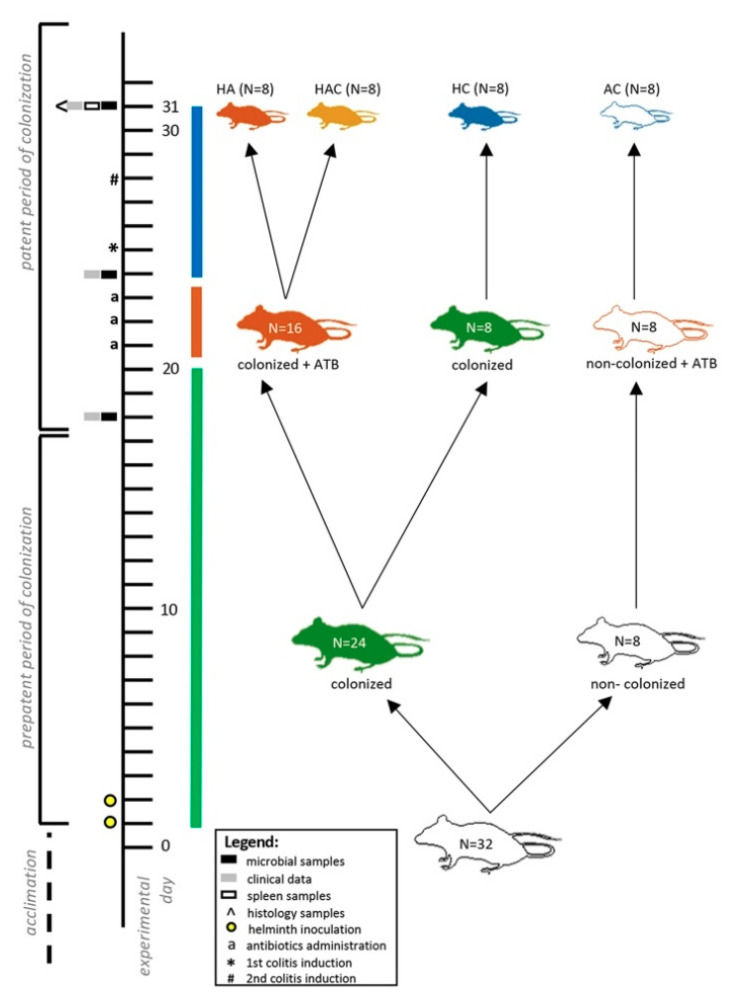
Graphic design of the experiment. The experiment involved three phases in which rats were affected by several factors: worm colonization (green bar; Days 1–20, but the *H. diminuta* colonization persisted throughout the experiment in the colonized rats), gut microbiota disruption (red bar; Days 21–24), and intestinal inflammation (blue bar; Days 25–31). The main monitoring and sample collection were performed on Day 31 (for more information see Material and Methods). There were four differently affected groups of rats: (i) treated with helminth and antibiotics (HA; red rat, N = 8); (ii) treated with helminth, antibiotics, and colitis (HAC; yellow rat, N = 8); (iii) treated with helminth and colitis (HC; blue rat, N = 8); (iv) treated with antibiotics and colitis (AC; white rat, N = 8). To evaluate the ongoing changes in the microbiota, fecal samples were collected throughout the experiment—on Days 0, 18, 25, and 31 (i.e., after colonization, after antibiotics administration, and after colitis induction). ATB—antibiotics.

**Figure 2 microorganisms-09-00226-f002:**
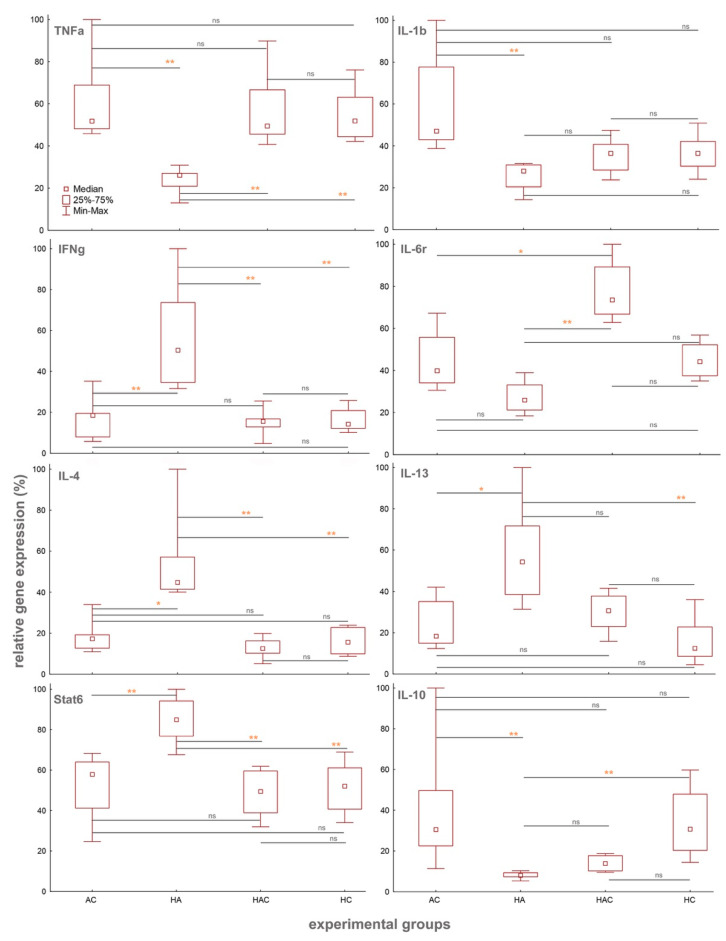
Changes in the relative gene expressions of all the monitored pro-inflammatory and anti-inflammatory markers (TNFα, IL-1β, IFNγ, IL-6r, IL-4, IL-13, Stat6, and IL-10) in the spleen samples. All cytokines were measured in all experimental groups of rats (AC, HA, HAC, and HC) on Day 31. All gene expressions are relative to the UBC housekeeping gene. Differences in the expressions of the genes among individual experimental groups were assessed by non-parametric Kruskal–Wallis ANOVA with multiple comparisons. ns—not significant; *—*p* < 0.05; **—*p* < 0.01.

**Figure 3 microorganisms-09-00226-f003:**
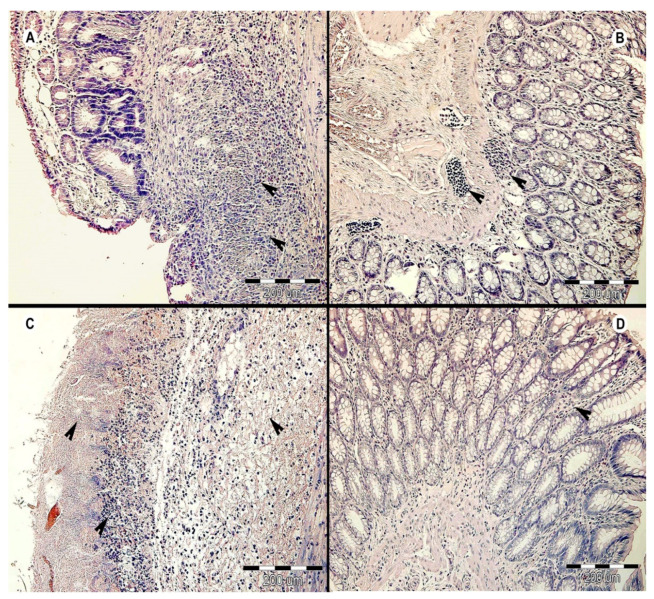
Histopathological examination of the large intestine of rats on Day 31 of the experiment. The representative cross sections are displayed from each experimental group: (**A**) AC group (i.e., antibiotic- and colitis-treated), black arrows indicate inflammatory infiltrate in mucosa and submucosa within necrotic tissue; (**B**) HA group (i.e., colonized and antibiotic-treated), black arrows indicate focal inflammatory infiltrate; (**C**) HC group (i.e., colonized and colitis-treated), black arrows—upper left indicates necrosis of mucosal layer, lower left indicates lymphoplasmacytic inflammatory infiltrate, and right indicates submucosal edema; (**D**) HAC group (i.e., colonized, antibiotic-, and colitis-treated), black arrow indicates inflammatory infiltrate.

**Figure 4 microorganisms-09-00226-f004:**
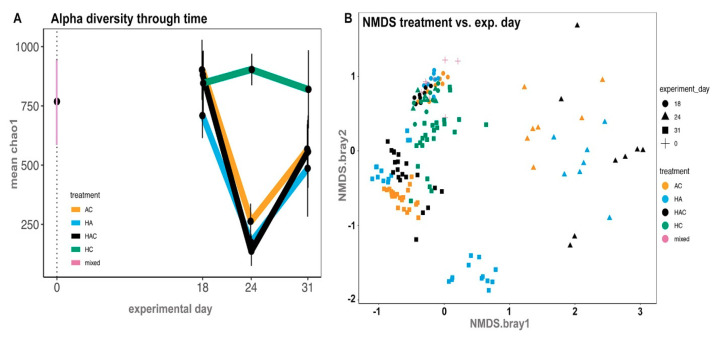
Microbial analyses throughout the experiment in rat fecal samples—after colonization (Day 18), after antibiotic treatment (Day 24) and after colitis treatment (Day 31). (**A**) Changes in the alpha diversity in the groups throughout the experiment; (**B**) NMDS analyses of the compositional changes in gut microbiota throughout the experiment, comparing treatment (in color) versus experimental day (in shape). For more details about the experimental groups and treatments, see Figure 1 or Appendix A.

**Figure 5 microorganisms-09-00226-f005:**
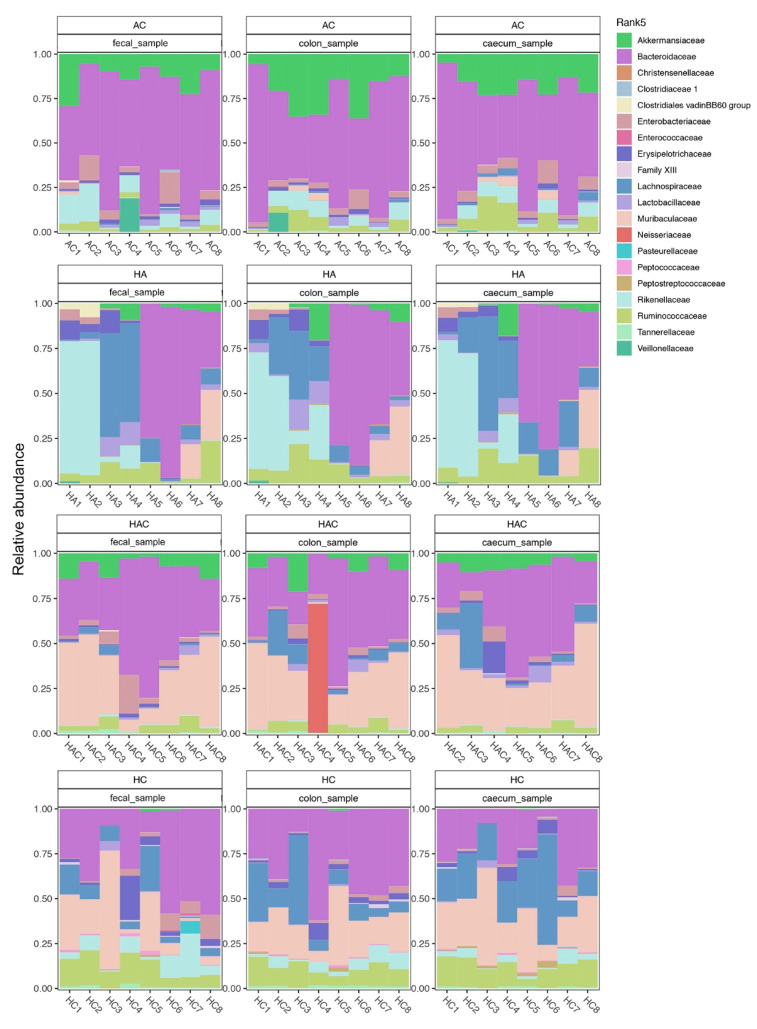
Summary of the bacterial taxa in the microbiota for each experimental group on Day 31. The microbial diversity and composition in each experimental group (AC, HA, HAC, and HC) is compared in three types of samples, specifically fecal samples, caecum, and colon. In each group and for each type of sample, the composition of the microbiota is shown in all individual rats. Rats were housed in pairs over the course of the experiment (i.e., nb.1 with nb.2, nb.3 with nb. 4, nb.5 with nb. 6, and nb.7 with nb. 8 in each group).

## Data Availability

The dataset of 16S sequences in this study can be found online through the European Nucleotide Archive, as described in the Material and Methods section.

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
