# Peer review of "Helminth Interactions with Bacteria in the Host Gut Are Essential for Its Immunomodulatory Effect"

_microorganisms, 2021, doi:10.3390/microorganisms9020226_

Round 1

Reviewer 1 Report

I congratulate the authors on the quality of the scientific work developed, a well-designed investigation, structured methodology, robust bibliographic research and correct scientific writing. Just a few points, I would like to comment (attach).

Reviewer 2 Report

The title is informative and relevant. The authors stated clearly what study found and how they did it. The references are relevant and recent. 

The study methods are valid and reliable. There are enough details provided in order to replicate the study.

They present the data appropriately. The text in the results adds to the data and it is not repetitive. Statistically significant results are apparent. Results are discussed from different angles and placed into context without being overinterpreted.

The conclusions are supported by references and own results. The study design is appropriate to answer the aim. This study added to what is already in the topic. The article is consistent within itself.

There are no major flaws in this article. However, the idea for use of helminths to modulate the immune system towards protection and tolerance instead of autoimmunity and cytotoxicity is well-established. However, up to date the clinical implication of these observations is not sufficiently effective. One might assume that the missing player in this interaction could be the host microbiome.

Specific comments on weaknesses of the article and what could be improved:

Major points  - none

Minor points

  1. It is not discussed enough which results are with practical meaning, especially in the clinical context. 
  2. The limitations of the study are not stated
  3. Figure 1 is missing
